# Clinical Features and Outcomes of Acute Kidney Injury in Critically Ill COVID-19 Patients: A Retrospective Observational Study

**DOI:** 10.3390/jcm12155127

**Published:** 2023-08-04

**Authors:** Nabil Bouguezzi, Imen Ben Saida, Radhouane Toumi, Khaoula Meddeb, Emna Ennouri, Amir Bedhiafi, Dhouha Hamdi, Mohamed Boussarsar

**Affiliations:** 1Faculty of Medicine of Sousse, University of Sousse, Sousse 4000, Tunisia; 2Medical Intensive Care Unit, Research Laboratory “Heart Failure”, LR12SP09, Farhat Hached University Hospital, Sousse 4000, Tunisia

**Keywords:** acute kidney injury, COVID-19, intensive care, outcomes, mortality

## Abstract

Background: An alarming number of COVID-19 patients, especially in severe cases, have developed acute kidney injury (AKI). Aim: The study aimed to assess the frequency, risk factors, and impact of AKI on mortality in critically ill COVID-19 patients. Methods: The study was a retrospective observational study conducted in the MICU. Univariate and multivariate analyses were performed to identify risk factors for AKI and clinical outcomes. Results: During the study period, 465 consecutive COVID-19 patients were admitted to the MICU. The patients’ characteristics were median age, 64 [54–71] years; median SAPSII, 31 [24–38]; and invasive mechanical ventilation (IMV), 244 (52.5%). The overall ICU mortality rate was 49%. Two hundred twenty-nine (49.2%) patients developed AKI. The factors independently associated with AKI were positive fluid balance (OR, 2.78; 95%CI [1.88–4.11]; *p* < 0.001), right heart failure (OR, 2.15; 95%CI [1.25–3.67]; *p* = 0.005), and IMV use (OR, 1.55; 95%CI [1.01–2.40]; *p* = 0.044). Among the AKI patients, multivariate analysis identified the following factors as independently associated with ICU mortality: age (OR, 1.05; 95%CI [1.02–1.09]; *p* = 0.012), IMV use (OR, 48.23; 95%CI [18.05–128.89]; *p* < 0.001), and septic shock (OR, 3.65; 95%CI [1.32–10.10]; *p* = 0.012). Conclusion: The present study revealed a high proportion of AKI among critically ill COVID-19 patients. This complication seems to be linked to a severe cardiopulmonary interaction and fluid balance management, thus accounting for a poor outcome.

## 1. Introduction

Severe Acute Respiratory Syndrome Coronavirus 2 (SARS-CoV-2) is a novel virus first detected in Wuhan, Hubei Province, China, in December 2019. This virus may cause severe viral pneumonia with acute respiratory distress syndrome causing millions of deaths all over the world. The 2019 Coronavirus disease (COVID-19) outbreak was declared an emerging pandemic with devastating consequences by the World Health Organization (WHO) in March 2020 [1]. Tunisia had more than 1,000,000 diagnosed cases and more than 28,000 deaths, and had the second-highest number of deaths in Africa [2]. The main feature of COVID-19 is lung involvement [3]. However, previous studies also reported the involvement of other organs (e.g., heart, liver, and kidney). In fact, an alarmingly high number of SARS-CoV-2 patients, particularly those with severe cases, have been documented to experience acute kidney injury (AKI) [4]. In the literature, the reported incidence of AKI varies widely. It is estimated to range between 6.5% and 46% in patients with COVID-19 infection [5,6], with the highest ranges in critically ill patients (23–81%) [7,8]. Data on AKI in Tunisian critically ill COVID-19 patients are scarce [9].

The present study aimed to assess the frequency, risk factors, and impact of AKI on ICU mortality in critically ill COVID-19 patients.

## 2. Materials and Methods

### 2.1. Study Design and Participants

This study was a retrospective observational study conducted in the Medical Intensive Care Unit (MICU) of Farhat Hached Teaching Hospital in Sousse, Tunisia. All COVID-19 adult patients admitted between March 2020 and December 2021 were included. The patients with pre-existing end-stage kidney disease (Grade 5 according to the 2012 KDIGO CKD defined as either a glomerular filtration rate less than 15 mL/min/1.73 m^2^ or a need for dialysis) were excluded [10]. To prevent sampling bias, consecutive sampling was used. The Research and Ethics Committee of Farhat Hached Teaching Hospital, Sousse, Tunisia, (IORG 0007439, Office for Human Research Protection—US Department of Health and Human Service) approved the study and waived the need for written informed consent as the study was a retrospective one (CER: 28-2022). The STROBE criteria have been fulfilled by this study (Appendix A).

### 2.2. Data Collection

Two intensivists who have received training in data handling collected all the data from medical charts in order to eliminate any potential interobserver and intraobserver bias. The following patients’ demographic and clinical characteristics were collected: age, gender, past medical history, Charlson comorbidity index (CCI) [11], Simplified Acute Physiology Score (SAPS II) [12], Sequential Organ Failure Assessment (SOFA) score [13], pre-ICU management delay and laboratory results (complete blood count, renal function, arterial oxygen partial pressure (PaO2)/fractional inspired oxygen (FiO2) (P/F ratio)), and inflammatory markers including C-reactive protein (CRP) and D-dimer. The use of vasopressors and ventilatory management with additional information on cumulative daily fluid balance was also documented. All admitted patients were divided into two groups: patients with AKI and without AKI; the patients with AKI were divided into two groups: survivors and non-survivors.

Outcomes collected were occurrence of AKI, Kidney Disease Improving Global Outcomes (KDIGO) stages [14], use of renal replacement therapy (RRT), fluid balance, use of invasive mechanical ventilation (IMV) and its duration, shock, ICU length of stay (LOS) and ICU mortality. There were no missing clinical data. Several missing lab results were noticed that reflect real-life intensivists’ diagnostic procedures.

### 2.3. Definitions and Variables

SARS-CoV-2 infection was determined as either a positive COVID-19 rapid antigen test or a positive Real-Time Reverse Transcriptase-Polymerase Chain Reaction (RT-PCR) assay of nasal swabs, combined with a high pre-test clinical probability. 

The Berlin criteria were used to assess the diagnosis and severity of the acute respiratory distress syndrome (ARDS) [15]. 

According to the World Health Organization’s reported criteria of illness classifications, the severity of COVID-19 disease was categorized as mild, moderate, severe, or critical (WHO) [16].

The Charlson comorbidity index (CCI) [11] is an assessment tool first developed by Charlson et al. in 1987 and used as a measure of comorbidity burden. It is a weighted index that takes into account the number and the seriousness of comorbid diseases to estimate the risk of death from comorbid conditions. 

The severity of illness was assessed by SAPS II and SOFA score based on the worst values collected during the first 24 h of ICU admission.

Post-intubation hypotension [17] was defined as a decrease in systolic blood pressure (SBP) to ≤90 mmHg, a decrease in SBP of ≥20% from a baseline, a decrease in mean arterial pressure to ≤65 mmHg, or the initiation of vasopressors within the 30 min following intubation.

The primary endpoint, AKI, was defined using the Kidney Disease Improving Global Outcomes (KDIGO) criteria, which include a change in serum creatinine of 0.3 mg/dL (26.5 mol/L) over a 48 h period, a 50% increase in baseline creatinine, or urine volume <0.5 mL/kg/h for 6 h [14]. The most recent serum creatinine value was taken into consideration as the baseline creatinine for individuals who had previous serum creatinine in the 7–365 days prior to admission [14]. In patients with suspected chronic kidney disease and for whom no baseline creatinine value was available, the lowest creatinine measured in the first three days of hospitalization was considered baseline serum creatinine. Finally, the patients with the highest serum creatinine levels at hospital admission, which declined thereafter during the ICU stay were considered as AKI.

Following KDIGO guidelines, the severity of AKI was divided into stages: (1) increase in serum creatinine of 0.3 mg/dL or increase to 1.5–1.9 times baseline serum creatinine or urine volume <0.5 mL/kg/h for 6 h; (2) increase to 2–2.9 times from baseline serum creatinine or urine volume <0.5 mL/kg/h for ≥12 h; (3) increase to more than three times baseline serum creatinine or a peak serum creatinine >4.0 mg/dL or urine output <0.3 mL/kg/h for ≥24 h or anuria for ≥12 h or if the patient received RRT during admission.

Daily fluid balance was calculated by subtracting the total fluid output (the sum of the volumes of urine output, ultrafiltration fluid, drain fluid, estimated gastrointestinal losses, and insensitive losses) from the total intake (the sum of all intravenous and oral fluids).

### 2.4. Statistical Analyses

For categorical variables, patient characteristics were expressed as frequencies and percentages; for continuous variables, as means and standard deviation; and for non-normal continuous variables, as medians and interquartile ranges. The Kolmogorov–Smirnov test was used to evaluate the distribution of the data. For comparing categorical variables, the chi-square or Fischer’s exact test was employed, and for comparing continuous variables, the Student’s t-test or Mann–Whitney U test were employed. Kaplan–Meier overall survival curves were produced and compared using log-rank testing until death or discharge. With the help of univariate and multivariate analyses, risk variables for ICU mortality were evaluated. False discovery rate analysis was not performed as the it was assumed that the sample size was sufficiently large to prevent such bias. 

## 3. Results

During the 22-month-long study period, a total of 732 patients were admitted to the ICU. In total, 465 (63.5%) patients had a confirmed COVID-19 infection among them and 229 (49.2%) patients developed AKI. The flowchart is displayed in Figure 1.

### 3.1. Baseline Characteristics

The main characteristics of the cohort are summarized in Table 1. The median age of the patients was 64 [54–71] years and 290 (62.4%) were male. Comorbidities were mainly hypertension (47.1%), diabetes mellitus (45.2%), obesity (38.2%), chronic kidney disease (11.4%), and coronary disease (10.8%). The median SAPS II score and SOFA score were 31 [24–38] and 3 [2–5], respectively. Overall, 244 (52.5%) patients received IMV, and 253 (54.4%) patients received vasopressor support.

### 3.2. Acute Kidney Injury

Two hundred twenty-nine (49.2%) patients developed AKI during their ICU stay with a mean onset from hospitalization at 4.59 ± 3.67 days. Among them, 30.1% had AKI within 48 h of ICU admission. Of all the AKI patients, 30 (13.1%) had AKI stage 1, 27 (11.8%) had AKI stage 2, 172 (75.1%) had AKI stage 3, and 47 (20.5%) received RRT.

### 3.3. Factors Associated with AKI

Table 1 displaying univariate analysis identified the following factors as associated with AKI; comorbidities (diabetes mellitus, obesity, and chronic kidney disease); CCI; severity scores (SOFA and SAPS); mean arterial pressure; leucocyte count; D-dimer; urea and CRP on admission; positive fluid balance and cumulative fluid balance at day 3; IMV; ICU LOS; shock during ICU stay; post-intubation hypotension; septic shock and right heart failure. The multivariate regression model identified the following factors as independently associated with AKI: positive fluid balance (OR, 2.78; 95%CI [1.88–4.11]; *p* = 0.001), right heart failure (OR, 2.15; 95% CI [1.25–3.67]; *p* = 0.005), and IMV (OR, 1.55; 95% CI [1.01–2.40]; *p* = 0.044).

Figure 2 displays the Kaplan–Meier plots of the respective IMV start, AKI onset, and the rate of positive fluid balance on days 1, 2, 3, 7, and 12. We can notice, early since day 2 of ICU stay, a diastasis of 1.35 days as for the difference respectively between IMV start and AKI onset after the ICU admission (3.24 ± 2.18 days and 4.59 ± 3.67 days). This 1.35-day difference could be interpreted as relatively short, but time is nephrons here, and IMV, as well as positive fluid balance, could have been rapidly accounted for, at least in part, by the AKI development and/or progression.

### 3.4. Predictors of ICU Mortality in AKI Patients

Among the 229 AKI patients, 135 (59%) died. A stepwise increase in the stage of AKI conferred an incremental risk of in-hospital death as shown in Figure 3. The ICU mortality within the AKI stages was, respectively, 43%, 44%, and 63.9%, in stages 1, 2, and 3.

In univariate analysis, age, comorbidities, CCI, SAPS II, stage 3 AKI, AKI mechanism (pre-renal vs. intrinsic), IMV, post-intubation hypotension, septic shock, and right heart failure were found to be associated (*p* < 0.05) with ICU mortality among AKI COVID-19 patients as shown in Table 2. Multivariate regression model identified the following factors as independently associated with ICU mortality among AKI COVID-19 patients: age (OR, 1.05; 95% CI [1.02–1.09]; *p* = 0.001), septic shock (OR, 3.65; 95% CI [1.32–10.10]; *p* = 0.012) and IMV (OR, 48.23; 95% CI [18.05–128.89]; *p* < 0.001).

## 4. Discussion

The present study revealed a relatively high proportion of AKI among critically ill COVID-19 patients. This complication seems to be linked to the severe cardiopulmonary interaction as assessed by positive fluid balance, IMV, and right heart failure; these were all surrogates of cardiac dysfunction and accounted for a poor outcome.

AKI in COVID-19 patients may be ascribed to different causes. SARS-CoV-2 uses angiotensin-converting enzyme 2 (ACE-2) as a receptor to facilitate viral entry into target cells. In fact, not only is it expressed in pulmonary type 2 alveolar cells, but also on the surface of kidney tubular cells. The infection of these cells may worsen the local inflammatory response (activation of macrophage, cytokine storm with high levels of interleukin 6 (IL-6), release of tissue factor, and activation of coagulation) leading to microcirculatory dysfunction, renal hypoperfusion, and tubular injury (viral-induced tubular or glomerular injury and thrombotic disease) [3,18,19,20]. Furthermore, hemodynamic instability and severe inflammation in critically ill patients may induce acute tubular necrosis (ATN) (sepsis-induced AKI) [19,21].

Regarding AKI incidence in the context of COVID-19, several previous studies have addressed this issue. Lowe et al. [22] found an AKI rate of 44% in critically ill patients. Ghosn et al. [23] reported an AKI incidence of 45.4%, while Lumlertgul et al. [24] showed that the incidence was even higher (76.6%) and the majority was in stage 3 (43.5%). Other reports showed AKI rates ranging from 18.3% to 57.4% in critically ill COVID-19 patients [3,7,25,26,27]. In the present study, AKI incidence was at 49%. The differences in the reported incidence rates of AKI might have several explanations. First, different definitions of AKI were used; in the present study, both serum creatinine and urine output criteria were adopted to define AKI, while other studies used only serum creatinine results. Second, differences in the severity of COVID-19 in patients may induce disparities in the incidence of AKI. Patients in the present study presented multiple comorbidities and were in quite severe condition. In fact, they had high severity scores. Additionally, 62.9% of them were mechanically ventilated and 57.6% needed vasopressors. These patients are more prone to develop AKI [28].

The proportion of AKI patients requiring renal replacement therapy (20.5%) found in this study was in line with other studies varying between 20.1% and 32% [7,8]. Determining the risk of developing AKI is a crucial step for the patient’s prognosis and the prompt implementation of preventive measures. Furthermore, AKI’s related consequences, such as long-term chronic renal disease or end-stage kidney disease, may be reduced with early diagnosis and appropriate treatment of AKI [29].

The present study included 465 subjects and explored the risk factors for AKI in critically ill patients with COVID-19. The factors associated with the development of AKI in the present study were IMV, positive fluid balance, and right heart failure. Interestingly, IMV had the same effect in COVID-19 patients as in other ICU patients [30]. This could serve as an example of the lung–kidney interaction theory because it relies on the effects of hypoxemia, a lack of decarboxylation, and the production of inflammatory cytokines as a result of ventilator-induced lung injury. Moreover, mechanical ventilation settings such as high levels of PEEP have direct effects on renal perfusion, especially when there is acute right ventricular failure [31,32].

Previous reports suggested that excessive fluid administration can be associated with the development of AKI [33]. Several mechanisms have been proposed to explain that [34]. First, fluid overload can induce a decrease in circulating intravascular volume secondary to capillary leakage as a consequence of endothelial dysfunction resulting from inflammation and ischemia–reperfusion injury. Second, excessive fluid administration can cause intrarenal compartment syndrome, interstitial edema, and venous congestion since kidneys are encapsulated organs. This could lead to increased resistance to venous return and can be responsible for renal ischemia [35,36]. Third, fluid overload can cause the stretching of atria and blood vessels and the activation of inflammatory cascades causing AKI. Fourth, massive fluid administration is a risk factor for abdominal compartment syndrome. Intra-abdominal hypertension may compress intra-abdominal vessels leading to increased renal venous pressure. This may lead to impaired renal plasma flow and a decrease in glomerular filtration. Finally, fluid overload increases bowel wall edema, compromises gut barrier function, and leads to bacterial translocation which can cause sepsis and AKI development [34]. 

Right heart failure is associated with a greater risk of AKI in the present study. Indeed, right ventricle dysfunction may affect renal function through pulmonary hypertension which has an inhibitory effect on left ventricular function [37] and may cause type 1 cardiorenal syndrome. In fact, right heart failure may lead to a decreased left ventricle cardiac output as a result of leftward bowing of the interventricular septum and central venous congestion [38]. This finding was reported by Chen et al. [39] who found that right ventricular dysfunction was associated with a significantly higher adjusted risk of AKI which tended to be more severe with right ventricular dysfunction than left ventricular dysfunction.

In the literature, other factors were reported to be associated with AKI in critically ill patients. Doher et al. [27] showed that the use of diuretics and creatinine levels at admission were independently associated with AKI. 

The ICU mortality rate in the present study was 59.2% which is in line with previous studies directed by Pineiro et al. in Spain (50%) [33], and Chan et al. in the USA (50%) [7]. Nonetheless, other studies conducted by Lumterttgul et al. [24] and Lowe et al. [22] showed a lower rate of 33% and 25%, respectively. The poor outcome of critically ill COVID-19 patients with AKI in the present study is probably due to the severity of illness scores in patients and the high rate of invasively ventilated patients. The risk factors associated with ICU mortality in COVID-19 patients with AKI were IMV use, septic shock, and age. Those factors have been already reported by Uchina et al. [40] in their multicenter study of 1738 patients. The results of the present study are in line with the findings of Bezerra et al. [41] who reported a death odd ratio of 8.44 for patients with AKI who underwent mechanical ventilation. According to Abumayyaleh et al. [42], sepsis in COVID-19 is associated with high mortality. In fact, it is designated as the main cause of death in COVID-19 patients, particularly if not diagnosed and treated promptly. Additionally, previous reports documented a higher frequency of hospital-acquired infections in critically ill COVID-19 patients. Septic AKI is associated with poor clinical outcomes [43] and an increased risk of death [44]. Early prediction and/or detection of septic shock, especially in AKI patients, can improve outcomes and prevent patient death [45].

An additional documented predictor for COVID-19 poor outcomes is advancing age. In a systematic review with meta-analyses conducted by Starke et al. [46], a high level of evidence showed a higher age-related risk of COVID-19 in-hospital lethality, case lethality, and admission of 5.7%, 7.4%, and 3.4% per age year, respectively. Furthermore, in line with the results of the present study, Marques et al. [47] found that older age was an independent risk factor associated with in-hospital mortality in COVID-19 AKI patients. In comparison to younger patients, the reserve capacity of older patients is low, resulting in a higher risk of death from acute medical events such as SARS-CoV-2 infection. Therefore, these elderly patients have a higher risk of developing sepsis and acute respiratory or heart failure [48]. 

The results of the present study are quite generalizable as they reflect the real-life ICU management of severe COVID-19-related ARDS patients highly at risk of AKI. Even independently of the cause of the ARDS, these results remain generalizable as a high proportion of AKI was pre-renal (29.2%), and within the intrinsic renal failure mechanisms (70.7%), acute tubular necrosis could have probably accounted for an important proportion of AKI along with specific viral direct renal injury, as the AKI onset was clearly delayed (4.59 ± 3.67 days from ICU admission).

As the relatively high proportion of AKI among critically ill COVID-19 patients, revealed by the present study, seems to be linked to the severe cardiopulmonary interaction, it is highly suggested to improve the management of hypoxemia, acute pulmonary hypertension, and right heart failure. This could be achieved by early goal-directed management intended to improve gas exchange, the lower work of breathing, and prevent P-SILI (Patient Self-Inflicted Lung Injury) [49] and VILI (Ventilator-Induced Lung Injury) [50], and patient discomfort and delirium [51] next to the management of hemodynamics, fluid balance, and cardiorenal interaction [52]. As the plateau pressure [53] and PEEP titration [54] and energy transferred to the respiratory system as monitored by mechanical power [55,56] are important targets to lung recruitment, intensivists should pay attention to right heart–lung interaction and its consequences on the kidney [39,55]. Precision medicine [57], using PV-tools [56], esophageal pressure, Electric Impedance Tomography [58,59,60,61], echocardiography, and EVLW (Extra-Vascular Lung Water) monitoring with the transpulmonary thermodilution technique [52] could be valuable decision-making tools to improve prognosis and prevent AKI. Implementation of specifically designed AKI bundles or machine learning models could improve AKI prediction and outcomes.

There are several potential limitations concerning the results of the present study.

First, it was a single-center and retrospective study but the data were collected prospectively and uniformly by trained residents with no clinical missing data. 

Second, in some cases with no available true baseline creatinine, using the lowest creatinine value within the first 3 days of hospitalization may have over- or underestimated the incidence of AKI. 

Third, one could argue that risk factors should happen early before the event of interest. In some patients, the identified risk factors do not necessarily occur before the onset of AKI, but it is worth remembering that COVID-19-related AKI, likely occurs as a result of a myriad of mechanisms and causes. In the present study, some patients who developed AKI prior to or early on at ICU admission probably accounted for factors other than those identified. These factors come thereafter within the first three days to add their contribution to the development or the progression of AKI. As for cumulative fluid balance, the choice of day 3 is dictated by the minimal clinically important difference principle and is intended to encourage intensivists to avoid positive fluid balance from day 1 rather than waiting until day 3 for its estimation. It is true that 30% of AKI occurred within the 48 h period (14.8% of the study population) but the cumulative fluid balance of the first 3 days could then be interesting to detect the 70% remaining AKI patients occurring in the 85.2% remaining patients within the studied population. Indeed, this proportion of patients with the early onset of AKI is the consequence of factors acquired before ICU admission. It remains naturally invisible to the risk factors, such as fluid balance, that will later settle in the ICU. As AKI is rather a dynamic event and not a static one, preventing its progression remains interesting all across the ICU length of stay. Another issue is that we choose fluid balance on day 3 to make our results readily comparable to previously performed studies [62,63]. The predictive factors we included in our model were merely selected to afford a dynamic predictive rule including early risk factors such as fluid balance and IMV, then evolutionary factors such as right heart failure which is reflecting the natural evolution and the phenomenology of COVID-19-related ARDS management. 

Fourth, urinalysis was not performed in all patients hampered by the burden of the pandemic. 

Fifth, there is also no histological material which limits the ability to assess the accurate mechanism of renal injury. 

Lastly, long-term outcomes were not assessed due to a short observation period. Hence, further research might help shed light on the long-term effect of COVID-19 on the kidneys.

## 5. Conclusions

The present study revealed a high proportion of AKI among critically ill COVID-19 patients. This complication seems to be linked to severe cardiopulmonary interaction and accounted for a poor outcome.

## Figures and Tables

**Figure 1 jcm-12-05127-f001:**
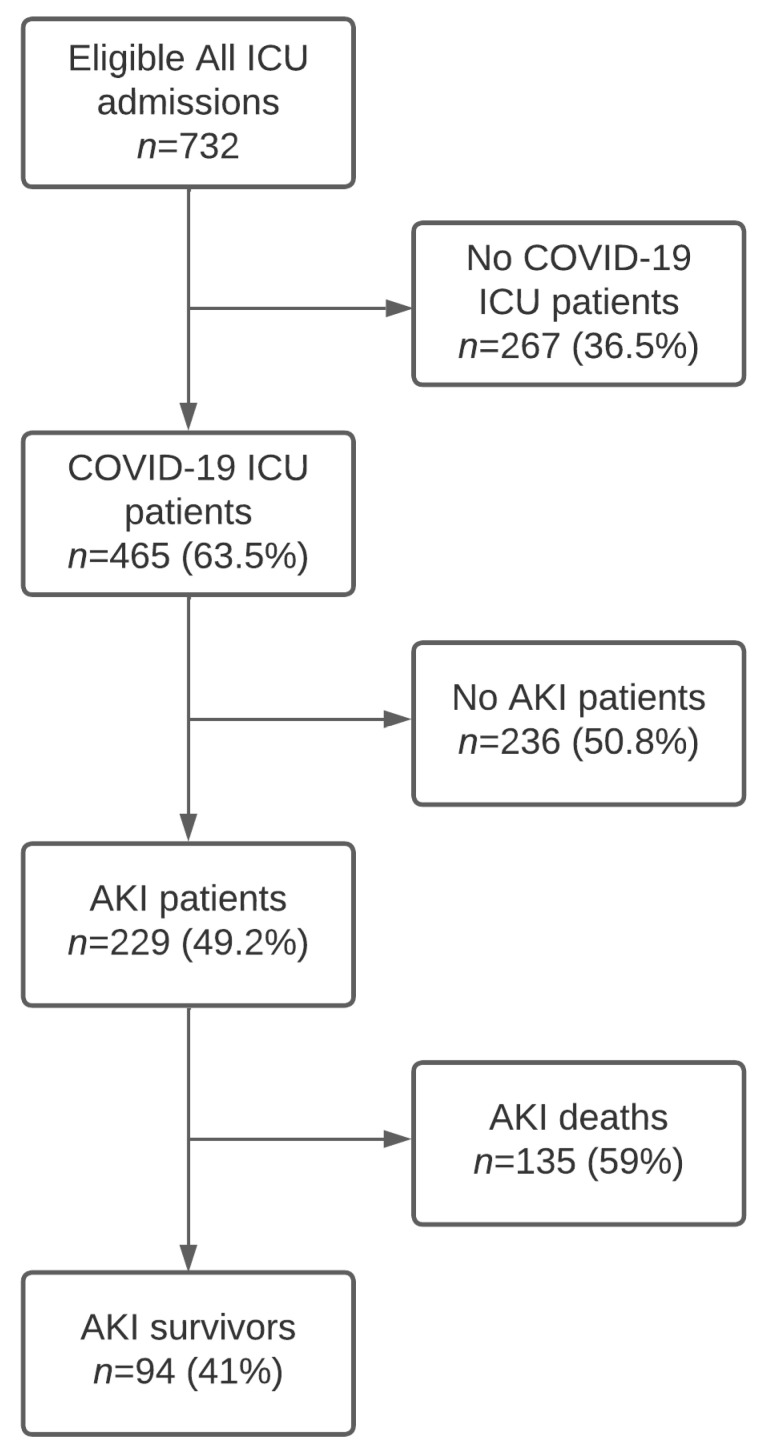
COVID-19 patient flow diagram.

**Figure 2 jcm-12-05127-f002:**
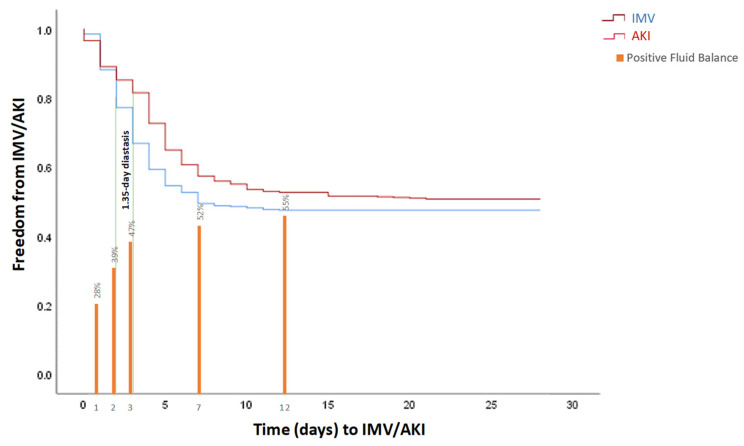
Kaplan–Meier estimates of freedom from IMV and AKI in critically ill COVID-19 patients with the rate of patients with positive fluid balance on days 1, 2, 3, 7, and 12. IMV: invasive mechanical ventilation, AKI: acute kidney Injury.

**Figure 3 jcm-12-05127-f003:**
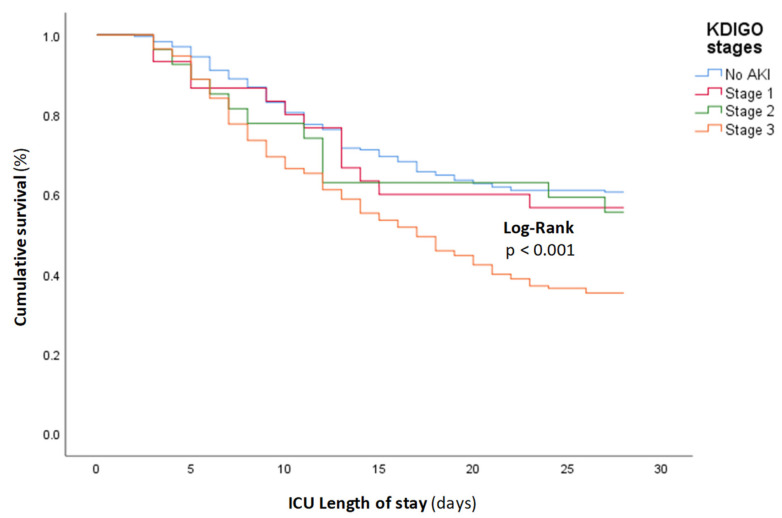
Kaplan–Meier curves for ICU mortality according to the stages of AKI.

**Table 1 jcm-12-05127-t001:** Comparison of patients’ demographics, clinical characteristics, management, and outcomes, by AKI status in COVID-19 patients.

	Total (*n* = 465)	AKI (*n* = 229)	No AKI (*n* = 236)	*p* Value
Age (years)	64 [54–71]	65 [55–71]	64 [53–71]	0.546
Male	290 (62.4)	137 (59.8)	151 (63.9)	0.356
CCI	3 [1–4]	3 [2–4]	2 [1–3]	<0.001
Comorbidities				
Hypertension	219 (47.1)	116 (50.7)	100 (42.4)	0.073
Diabetes	210 (45.2)	123 (53.7)	86 (36.4)	<0.001
Obesity	178 (38.2)	106 (46.3)	72 (30.5)	<0.001
Coronary disease	50 (10.8)	19 (8.3)	29 (12.3)	0.157
Chronic kidney disease	53 (11.4)	35 (15.3)	18 (7.6)	0.009
Chronic pulmonary disease	17 (3.7)	7 (3.1)	12 (5.1)	0.269
Neoplasia	13 (2.8)	6 (2.6)	6 (2.5)	0.958
Immunosuppression	29 (6.2)	18 (7.9)	10 (4.2)	0.101
Illness severity scores				
SOFA	3 [2–5]	4 [3–5]	3 [2–4]	<0.001
SAPS II	31 [24–38]	33 [27–42]	29 [23–35]	<0.001
Pre-ICU management delay (days)	11 [8–14]	11 [8–14]	11 [8–14]	0.307
Intensive care unit course				
IMV on admission	14 (3)	10 (4.4)	4 (1.7)	0.123
Shock on admission	35 (7.5)	25 (10.9)	10 (4.3)	0.007
P/F ratio at admission	120 [90–160]	120 [84–160]	118 [85–160]	0.287
Use of invasive ventilation	244 (52.5)	144 (62.9)	100 (42.4)	<0.001
Duration of IMV, (days)	7 [4–11]	8 [4–12]	7 [4–12]	0.422
Use of vasopressors	253 (54.4)	150 (65.5)	103 (43.6)	<0.001
Duration of vasopressor use, (days)	7 [4–11]	7 [5–12]	7 [4–12]	0.074
Positive fluid balance on day 3	219 (47.1)	139 (60.7)	80 (34.3)	<0.001
Cumulative fluid balance at day 3, (mL)	−135 [−2000/+2162]	690 [−1275/+3275]	−550 [−2450/+800]	<0.001
ICU LOS, (days)	8 [5–13]	9 [6–16]	7.5 [5–12]	0.001
Adverse events				
Shock during ICU stay	212 (45.5)	132 (57.6)	80 (33.9)	<0.001
Post-intubation hypotension	93 (20)	58 (25.3)	35 (14.8)	0.005
Septic shock	143 (30.8)	87 (38)	56 (23.7)	0.001
Right heart failure	103 (22.2)	71 (31)	32 (13.6)	<0.001
Hemorrhagic shock	9 (1.9)	4 (1.7)	5 (2.1)	-
ICU mortality	228 (49)	135 (59.2)	93 (40.8)	<0.001
Admission laboratory profile				
Creatinine (µmol/L), (*n* = 420)	70 [54–96]	70 [57–100]	69 [57–89]	0.499
Urea (mmol/L), (*n* = 420)	7 [5–10.5]	7.7 [5–11]	7 [5–10]	0.014
CRP (mg/L), (*n* = 109)	130 [74–176]	154 [116–238]	112 [62–162]	0.005
WBC (10^9^/L), (*n* = 328)	10.3 [7.7–14.6]	11.2 [7.9–15.1]	9.8 [7.4–14]	0.167
Hb (g/dL), (*n* = 328)	12 [10.2–13]	11.8 [10–12.8]	12 [10.8–13]	0.090
Lymphocytes (10^9^/L), (*n* = 328)	0.7 [0.5–1]	0.7 [0.5–1]	0.8 [0.5–1]	0.737
D-dimer (µg/mL), (*n* = 49)	1000 [507–2234]	1300 [511–2238]	838 [506–1921]	0.244
NT-proBNP (pg/mL), (*n* = 93)	664 [238–2734]	410 [224–2111]	930 [414–3280]	0.429
pH, (*n* = 393)	7.43 [7.36–7.46]	7.42 [7.36–7.46]	7.43 [7.37–7.46]	0.276
Bicarbonates (mmol/L), (*n* = 393)	23 [20–25]	23 [20–25]	23.6 [21–26]	0.795

Data were expressed as *n* (%) for categorical variables and as median [IQR] or mean ± SD for continuous variables. Probability (*p*): comparison between the 2 groups (Wilcoxon–Mann–Whitney test for continuous data and chi-square test for categorical data). AKI: acute kidney injury, CCI: Charlson comorbidity index, COPD: chronic obstructive pulmonary disease, SAPS II, Simplified Acute Physiological Score II, SOFA: Sequential Organ Failure Assessment, ICU: intensive care unit, IMV: invasive mechanical ventilation, P/F ratio: the arterial partial pressure of oxygen (PaO2) divided by the inspired oxygen concentration (FiO2), Hb: hemoglobin, WBC: white blood cells, LOS: length of stay.

**Table 2 jcm-12-05127-t002:** Factors associated with ICU mortality in AKI COVID-19 patients.

	All AKI Patients (*n* = 229)	Survivors (*n* = 94)	Non Survivors (*n* = 135)	*p* Value
Age, (years)	65 [55–71]	58 [47–67]	68 [62–73]	<0.001
Male	137 (59.8)	51 (54.3)	86 (63.7)	0.151
CCI	3 [2–4]	2 [1–4]	3 [2–4]	0.009
Comorbidities				
Hypertension	116 (50.7)	40 (42.6)	76 (56.3)	0.041
Diabetes	123 (53.7)	41 (43.6)	82 (60.7)	0.011
Obesity	106 (46.3)	35 (37.2)	71 (52.6)	0.022
Chronic pulmonary disease	7 (3.1)	2 (2.1)	5 (3.7)	0.496
Chronic kidney disease	35 (15.3)	9 (9.6)	26 (19.3)	0.045
Neoplasia	6 (2.6)	0 (0)	6 (4.4)	-
Immunosuppression	18 (7.9)	6 (6.4)	12 (8.9)	0.488
Illness severity scores				
SOFA	4 [3–5]	4 [2–5]	4 [3–5]	0.179
SAPS II	33 [27–42]	27.5 [24–38.5]	34 [29–44]	0.001
KDIGO stages				
Stage 1	30 (13.1)	17 (18.1)	13 (9.6)	0.062
Stage 2	27 (11.8)	15 (16)	12 (8.9)	0.103
Stage 3	172 (75.1)	62 (66)	110 (81.5)	0.008
Etiology of AKI				
Prerenal	67 (29.2)	17 (18.1)	50 (37)	0.002
Type 1 CRS	37 (16.2)	0	37 (27.4)	-
Other than CRS	30 (13.1)	17 (18.1)	13 (9.6)	-
Intrinsic renal	162 (70.7)	77 (81.9)	85 (63)	0.002
ICU course				
IMV use	144 (62.9)	17 (18.1)	127 (94.1)	<0.001
Duration of IMV (days)	8 [4–12]	6 [3–9]	8 [4–12]	0.426
RRT	47 (20.5)	3 (3.2)	44 (32.6)	<0.001
Cumulative fluid balance at day 3 (mL/3 days)	690 [−1275/+3275]	750 [−1275/+3750]	650 [−1350/+3000]	0.618
Positive fluid balance	139 (60.7)	56 (59.6)	83 (61.5)	0.771
ICU LOS, (days)	9 [6–16]	8 [5–13]	10 [7–17]	0.185
Adverse events				
Shock	132 (57.6)	9 (9.6)	123 (91.1)	<0.001
Post-intubation hypotension	58 (25.3)	16 (17)	42 (31.1)	0.016
Septic shock	87 (38)	9 (9.6)	78 (57.8)	<0.001
Right heart failure	71 (31)	0	71 (52.6)	-
Hemorrhagic shock	4 (1.7)	0	4 (3)	-

Data were expressed as *n* (%) for categorical variables and as median [IQR] or mean ± SD for continuous variables. Probability (p): comparison between the 2 groups (Wilcoxon–Mann–Whitney test for continuous data and chi-square test for categorical data). AKI: acute kidney injury, CCI: Charlson comorbidity index, COPD: chronic obstructive pulmonary disease, SAPS II: Simplified Acute Physiological Score II, SOFA: Sequential Organ Failure Assessment, CRS: cardiorenal syndrome, IMV: Invasive Mechanical Ventilation, RRT: renal replacement therapy, ICU: intensive care unit, LOS: length of stay.

## Data Availability

The datasets generated and/or analyzed during the current study are not publicly available due to limitations of ethical approval involving the participants’ data and anonymity but are available from the corresponding author on reasonable request.

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
