# Peer review of "Clinical Features and Outcomes of Acute Kidney Injury in Critically Ill COVID-19 Patients: A Retrospective Observational Study"

_jcm, 2023, doi:10.3390/jcm12155127_

Round 1

Reviewer 1 Report

The presented research describes the incidence of AKI in patients with a severe form of COVID-19 disease. Chronic kidney disease is mentioned as an exclusion criterion. However, it is included in Tables 1 and 2. How did you diagnose AKI in patients with CKD?

The diagnosis of the postintubation shock does not exist. Patients can develop hypotension following the intubation, but not shock. Moreover, there are tools that enable the prediction of hypotension development and physicians' duty is to try to prevent it. The above-mentioned are serious flaws in the terminology and methodology of the current work.

Reviewer 2 Report

"Clinical Features and Outcomes of Acute Kidney Injury in Critically Ill COVID-19 Patients, A Retrospective Observational Study," by Drs. Boussarsar et al. describe 229 patients with acute kidney injury who presented to the Farhat Hached University Hospital Medical Intensive Care Unit out of 465 patients infected with  severe acute respiratory syndrome coronavirus-2 infections. Analysis (multivariate) of these individuals suggested that fluid balance, right heart failure, and mechanical ventilation were associated with renal insufficiency in this cohort. The writing was clear, methods straightforward, and the results support the conclusions. I have a few suggestions.

Line 140. what test was used for statistical comparison? Multiple tests are mentioned in Statistical Analyses (line 106). Also, it should be noted that there was no false discovery rate analysis.  

Abstract. 

Methods. Lines 96-101. The list should numerate the AKI stages by something other than a right slash (which suggests division). E.g., "...AKI was divided into stages: 1), ... 2), ... 3),..."

Results. 

Figure 3. The legend font (KDIGO stage) is too small even in PDF format. This needs to be increased. 

Discussion. Although the points delivered are clear to this reviewer, there are a few grammatical errors. Use a grammar-checking software for the Discussion section for the next revision. 

Reviewer 3 Report

Dear authors,

AKI in critically ill covid-19 patients is an interesting issue that has not been studied extensively. I read with interest your manuscript about the clinical features and outcomes of AKI, although this is not a novel material in the literature.

Discussion: Please add citations to lines 175-178. Please comment on the novel info provided by this study. Do any suggestive protecting measures arise based on the study?

Methods: English check for minor corrections.

Round 2

Reviewer 1 Report

Dear authors, thank you for answering some questions and revising the manuscript. However, you still did not explain how you diagnosed AKI in patients with already-known CKD.

Author Response

See attached doc.
